# Assessment of the Impact of Ceftriaxone on the Functional Profile of Soil Microbiota Using Biolog EcoPlate<sup>TM</sup>

Livia da Silva Freitas [1,2,3], Rodrigo de Lima Brum [1,3], Alícia da Silva Bonifácio [1,3], Lisiane Martins Volcão [1,3], Flavio Manoel Rodrigues da Silva Júnior [1,3,4,*] and Daniela Fernandes Ramos [1,2]

1    Programa de Pós graduação em Ciências da Saúde, Faculdade de Medicina, Universidade Federal do Rio Grande-FURG, Rio Grande 96200-000, Brazil
2    Laboratório de Desenvolvimento de Novos Fármacos (LADEFA), Faculdade de Medicina, Universidade Federal do Rio Grande-FURG, Rio Grande 96200-000, Brazil
3    Laboratório de Ensaios Farmacológicos e Toxicológicos, Instituto de Ciências Biológicas, Campus Carreiros, Universidade Federal do Rio Grande-FURG, Rio Grande 96203-900, Brazil
4    Faculdade de Medicina, Universidade Federal do Rio Grande-FURG, Rio Grande 96200-000, Brazil
*    Correspondence: f.m.r.silvajunior@gmail.com; Tel.: +55-53-32374634

**Abstract: Background:** Antibiotics are essential to the treatment of diseases, but they have also brought about concerns in terms of their environmental, economic, and health impacts. Antibiotics can be excreted in unchanged form or as metabolites, which can cause toxicity by contaminating different environmental compartments, including soil. Soil is a critical compartment due to the numerous functions it performs and its direct impact on the communities of microorganisms, plants, and animals that make up the soil ecosystem. The functional profile of soil microbiota has emerged as a promising tool to assess soil quality. This study aimed to evaluate the functional profile of soil microbiota and the gut microbiota of earthworms in ceftriaxone-contaminated soil using Biolog EcoPlate. **Methods:** Soil samples contaminated with varying concentrations of ceftriaxone (0, 1, and 10 mg/kg) were incubated for 14 days in the presence or absence of the earthworm *Eisenia andrei*. After exposure, the physiological profile of the soil microbiota and the gut microbiota of the earthworms were evaluated using Biolog EcoPlate. **Results:** No significant differences were observed in the parameters evaluated using different concentrations of the antibiotic. The functional profile of the microbiota in the soil with and without earthworms was found to be similar, but interestingly, it differed from the profile of the intestinal microbiota of the earthworms. **Conclusions:** The findings of this study indicate that the presence of earthworms did not significantly alter the functional profile of the soil microbiota in ceftriaxone-contaminated soil. Further studies are necessary to investigate the potential impact of ceftriaxone and other antibiotics on soil microbiota and the role of earthworms in this regard.

**Keywords:** antibiotics; ceftriaxone; ecotoxicity; earthworms; *Eisenia andrei*

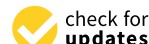



## 1. Introduction

Pharmaceutical products represent a significant milestone in the field of science, saving lives, extending life expectancy, treating illnesses, enhancing well-being, and improving the quality of life. The consumption of pharmaceutical products by humans and animals encompasses antibiotics, hormones, and anti-inflammatories [1]. Antibiotics, in particular, hold indisputable importance, as they have been widely employed in human and veterinary medicine for decades to combat diseases and infections, increase production, and boost agroindustrial performance [2]. However, these compounds have raised numerous environmental, economic, and health-related concerns. Typically, approximately 50–60% of these compounds are excreted unchanged in urine, leading to the contamination of diverse environmental compartments [3]. In addition to environmental contamination, the surge in antimicrobial resistance poses another critical health and economic concern. As per

the World Health Organization, the number of deaths related to antimicrobial resistance could climb up to 10 million people by 2050, while the economic losses could reach up to USD 10 trillion annually [4].

During the COVID-19 pandemic, there has been an upsurge in the utilization of antibiotics, sometimes inappropriately and without bacterial infection evidence [5]. Cephalosporins, a widely used antibiotic class, are beta-lactams, along with carbapenems, penicillins, and monobactams. The beta-lactam ring in their chemical structure sets beta-lactams apart from other antibiotics, inhibiting bacterial peptidoglycan cell wall synthesis and leading to prokaryotic cell lysis [6]. Due to their low toxicity and high therapeutic efficacy [7], they are a crucial antibiotic category. Cephalosporins are the most widely prescribed antibiotics worldwide, with numerous studies showing their broad-spectrum activity and potent bactericidal effect [8–10]. In livestock farming, they are used to treat bacterial infections in pigs, cattle, sheep, and poultry [11]. Moreover, cephalosporins are the antibiotics most prescribed and used by veterinarians globally [12,13]. Third-generation Ceftiofur is the most consumed cephalosporin in veterinary medicine, with more than 260 tons used in China alone in 2018 [14]. In this context, given the rising use of antibiotics, there is an extensive discussion regarding the potential consequences of their introduction into the environment. It is well established that antibiotics can alter the functional, structural, and genetic diversity of microbial communities, ultimately promoting the selection of antibiotic-resistant microorganisms. Conversely, recent evidence suggests that certain microorganisms may adapt to the presence of these compounds and convert them into less toxic products, thereby facilitating the restoration of the original microbiota within that environmental compartment [15].

Soil is a crucial environmental compartment that performs multiple functions, including food production, nutrient cycling, water regulation and purification, carbon and greenhouse-gas sequestration, habitat and biodiversity maintenance, organic matter decomposition, and water table recharge when well managed [16]. Moreover, soil hosts a diverse array of microorganisms that comprise the soil microbiota. Previously, studies were primarily focused on the fundamental role of soil in the nitrogen cycle. However, recently, it has gained even more attention due to being recognized as one of the key components in global carbon cycling, primarily due to the microbial communities that constitute it as an ecological matrix [17]. This has profound implications in the complexity of this ecosystem, which consists of microorganisms, plants, and diverse fauna, particularly invertebrates, that play a pivotal role in maintaining the physicochemical and microbial properties of the soil [18]. Furthermore, the abiotic soil environment is highly heterogeneous and exerts direct or indirect influences on the functional profile of soil, as well as on the communities of microorganisms, plants, and animals that it harbors [19]. Assessing the diversity of soil microbial communities is a crucial way to evaluate soil quality and health. Any biotic or abiotic disturbance can affect fundamental soil functions, such as nutrient cycling, biomass production, biogeochemical cycling, and soil formation. Among the sensitive indicators of the microbial response to antibiotic stress in soil are some techniques that include the enzymatic activity of enzymes such as dehydrogenase, phosphatase, and urease [20–22]. Biolog EcoPlate™ (Biolog, Inc., Hayward, CA, USA) is a widely used technique that allows the in-depth analysis of the metabolic profile of microbial communities to be performed [23–25]. Some organisms also play a significant role in maintaining soil ecosystem processes; invertebrates, such as earthworms, have been proposed to cause significant microbial changes in soil [26,27]. This condition has been exploited in industries that aim to promote the biodegradation of organic pollutants, including antibiotics, and the stabilization of inorganic pollutants, such as metals [28–30]. The combined impact of abiotic and biotic factors influences the biodiversity of environmental matrices, such as soil.

Soil quality can be measured with the physiological diversity of the microbiota and even the transfer of microorganisms through cultivated food to the gut microbiota of animals and humans. Several authors have proposed that the diversity of soil microbial communities is directly proportional to the resistance and resilience of this ecological

matrix to environmental disturbances [31,32]. In other words, the presence of contaminating compounds such as antimicrobials may result in changes in soil homeostasis and, consequently, in its ecological functionality [15]. The change in the physiological status of microbiota mediated by the addition of chemical contaminants can compromise ecosystem processes, including nutrient cycling. Antibiotics are particularly interesting to be studied in this context, since they have microorganisms as a therapeutic target. Previous evidence already points to physiological changes mediated by microbial exposure to antibiotics [33], including the intestinal microbiota of invertebrates [34]. Thus, the objective of the present study was to evaluate the impact of ceftriaxone contamination on the functional profile of soil microbiota in the presence and absence of the earthworm species *Eisenia andrei*.

## 2. Materials and Methods

### 2.1. Soil Sampling

Soil samples were collected from a protected area located on the campus of Federal University of Rio Grande (FURG) at the geographic coordinates of 32°04′37.4″ S 52°10′06.9″ W. This area has been used as a control in previous studies due to its low level of contamination [35–37]. Samples were collected using a shovel at a depth of 10 to 15 cm, and any plant residues were removed. The soil was then placed in plastic boxes and air-dried at room temperature for approximately four days. Subsequently, the soil was sieved using a 2 mm mesh and distributed into experimental containers at a weight of 250 g per container.

### 2.2. Test Organism and Assessed Chemical

The earthworm species Eisenia andrei, raised at Laboratory for Pharmacological and Toxicological Testing, was used in the experiments. The tested compound was 100% pure ceftriaxone (third-generation cephalosporin) obtained from Sigma Aldrich and was resuspended in sterile distilled water. The concentrations of 1 mg/kg and 10 mg/kg used in the experiments were based on a previous study by Orlewska and colleagues [38].

### 2.3. Experimental Design

The experiment consisted of two experimental groups: soil contaminated with the antibiotic in the presence and absence of earthworms. In treatments with earthworms, 10 clitellate earthworms were added to each container. Sterile plastic containers with a volume of 500 mL, measuring 142 mm in length, 98 mm in width, and 47 mm in height, were filled with 250 g of soil prepared as previously described. Then, 50 mL of antibiotic solution or sterile distilled water (equivalent to 50% of the soil's field capacity) was added in triplicate. The experimental groups were negative control, 1 mg/kg, and 10 mg/kg of Ceftriaxone, with and without earthworms. The experiment was conducted in triplicate and lasted for 14 days at 25 °C under a 12 h/12 h light/dark photoperiod. During the exposure period, the earthworms were not fed; the humidity of each replicate was monitored weekly; and no water replacement was necessary.

### 2.4. Microbial Functional Profile

The microbial functional profile was assessed using the EcoPlate system (Biolog, Inc., Hayward, CA, USA), which is a widely used tool in ecotoxicological assays, after 14 days of exposure [33–39]. This system measures the utilization of 31 different carbon sources that are associated with microorganisms present in the evaluated matrix. These carbon sources are divided into six substrate categories: carboxylic acids, polymers, amino acids, carbohydrates, amines/amides, and control (well without substrate), as presented in Table 1. The comparison among groups was made based on the amount of substrate utilized.

**Table 1.** List of substrates evaluated with the functional profile evaluation assay using the EcoPlate system, which uses only water as control. The substrates present in EcoPlate belong to five groups of compounds that can be oxidized to carbon (polymers, carbohydrates, amino acids, carboxylic acids, and amines/amides).

| Substrates | Carbon Sources |
| --- | --- |
| Polymers | Tween 40 |
| | Tween 80 |
| | $\alpha$-Cyclodextrin |
| | Glycogen |
| Carbohydrates | D-Cellobiose |
| | $\alpha$-D-Lactose |
| | beta–Methyl-D-Glucoside |
| | D-Xylose |
| | i-Erythritol |
| | D-Mannitol |
| | N-Acety l-D-Glucosamine |
| | Glucose-1-Phosphate |
| | D, L-$\alpha$-glycerol Phosphate |
| | D-Galactonic acid $\gamma$-Lactone |
| Carboxylic acids | Pyruvic acid methyl ester |
| | D-Glucosaminic acid |
| | D-Galacturonic acid |
| | $\gamma$-Hydroxybutyric acid |
| | Itaconic acid |
| | $\alpha$-Ketobutyric acid |
| | D-Malic acid |
| Amino acids | L-Arginine |
| | L-Asparagine |
| | L-Phenylalanine |
| | L-Serine |
| | L-Threonine |
| | Glycyl-L-glutamic acid |
| | 2-Hydroxy benzoic acid |
| | 4-Hydroxy benzoic acid |
| Amines/amides | Phenylethylanine |
| | Putrescine |

Samples of 5 g of soil were collected from each experimental group and mixed with 45 mL of 0.85% NaCl solution in a Falcon tube. Afterward, a dilution (150×) was performed in saline solution; then, 150 μL of each suspension was added in triplicate to each well of the plate [40]. For groups containing earthworms, we also analyzed the intestinal contents of these organisms. The earthworms were separated from the soil, washed with 70% alcohol, and dissected. The intestinal contents of each worm pool present in each group were transferred to Falcon tubes and macerated with three glass beads in 3 mL of 0.85% NaCl solution under stirring. A dilution (10×) was performed; then, 150 μL of each suspension was added in triplicate to each well of the plate. The plates were incubated in a DBO-type incubator at Laboratory of Pharmacological and Toxicological Assays (LEFT) at 28 °C. Readings were taken at 590 nm (FilterMax F5) after 48 h, 72 h, 96 h, and 120 h. The utilization rates of these carbon compounds were quantified according to the color change resulting from the transformation of soluble triphenyl tetrazolium chloride present on the plates, which changes to the reduced state: formazan. After each reading, the plates were re-incubated until the end of 120 h.

*2.5. Analysis of EcoPlate Data*

The average well color development and the Shannon diversity index (H) were derived with calculations based on absorbance measurements. The Shannon diversity index was computed using the following formula:

$$\text{'H'} = -\sum[(\text{pi}) \times \ln(\text{pi})]$$

where pi = ni/N, ni is the number of individuals of species i, and N is the total number of individuals. This is a relationship between abundance and richness expressing the uniformity of abundance values across all species in the sample.

The Shannon diversity index (H) is utilized to assess the physiological diversity (functional diversity) of bacterial communities. According to Muñiz et al. [41], microbial communities capable of degrading a greater number of substrates and/or demonstrating comparable degradation efficiency exhibit higher H values when compared with the metabolically inactive portion of the community that is unable to grow under plate conditions. For evaluating the overall capacity of the microbiota to utilize diverse carbon sources, the average well color development (AWCD) was individually determined for each incubation time employing the following equation:

$$\text{AWCD} = [\Sigma\,(C - R)]/n$$

where C represents the absorbance value of the control wells (averaged over 3 controls), R denotes the average absorbance of the response wells (3 wells per carbon substrate), and n corresponds to the number of carbon substrates (n = 31). AWCD serves as an indicator that reflects the comprehensive metabolic activity potential of the microbial community, thereby serving as a total bioactivity index for Biolog plates, as documented in previous studies [42,43]. Two other indices were also evaluated: NUSE and PUSE. These indices are associated with the use of nitrogen- and phosphorus-containing carbon sources, respectively. Both are represented by the percentage of the sum of the absorbance values from sources containing nitrogen or phosphorus in relation to the sum of the 31 absorbance values contained in each microplate replicate. Among the 31 substrate sources allocated to Biolog EcoPlate[TM], 10 included nitrogen (8 amino acids, amines, and amides), while 2 included phosphorus (both present in the carbohydrate group), and the rest of the 19 sources were mainly composed of carbon [44].

*2.6. Statistical Analysis*

The average well color development, Shannon index, NUSE, and PUSE values were consolidated into a single measurement by calculating the area under the curve, taking into account the incubation times and treatments. Subsequently, a two-way ANOVA test was conducted using GraphPad Prism version 8 software to assess the differences in the AWCD, Shannon index, NUSE, and PUSE values, considering the antibiotic concentrations and exposure times. Multivariate permutation analysis, two-way PERMANOVA, was employed to assess the differences in carbon source consumption among the treatments (control, 1 mg/kg, and 10 mg/kg) and groups (soil, soil with earthworms, and earthworm gut) in the 120-h time interval. All analyses were performed considering a significance level of $p < 0.05$.

**3. Results**

In the treatments where earthworms were present, no mortality was observed. Figure 1 illustrates the temporal dynamics of the Shannon diversity index during the four monitored incubation days (48 to 120 h) of Biolog EcoPlate, depicting the soil microbiota without and with earthworms (Figure 1a,b, respectively), as well as the earthworm gut microbiota (Figure 1c). Overall, the behavior of the two treatments involving the antibiotic ceftriaxone was comparable to the control, and this trend was consistent across the other assessed parameters (Figures 2–4). Figure 5 presents the mean areas under the curve of the four

examined parameters (Shannon diversity index, average well color development, NUSE, and PUSE) considering the comparison of two factors: ceftriaxone concentration in the soil and the source of the microbiota. Except for the Shannon diversity index, no significant differences were observed in the evaluated factors nor in the interaction among them (Table 2). Regarding the Shannon diversity index (Figure 5a), the functional profile of the earthworm gut microbiota exhibited statistically significant differences compared with the soil microbiota in the presence or absence of earthworms.

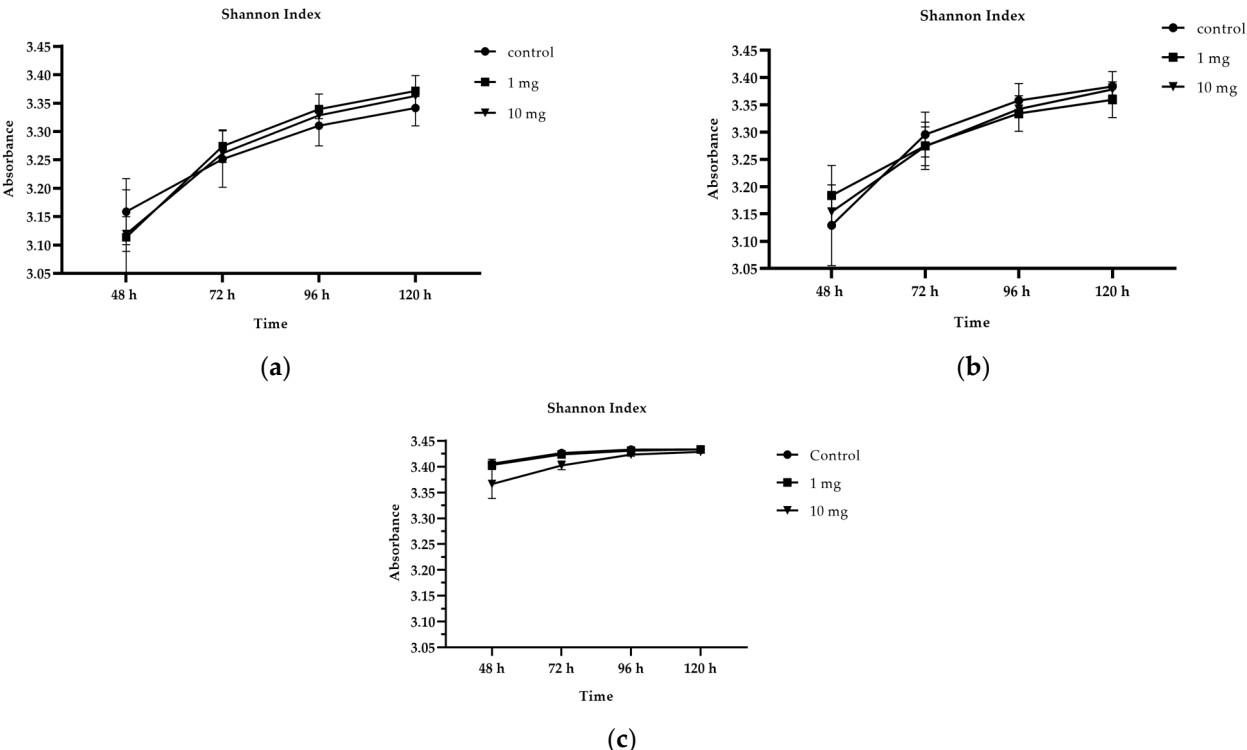

**Figure 1.** Shannon index of (**a**) soil, (**b**) soil with earthworms, and (**c**) earthworm gut under the three treatments (control, 1 mg, and 10 mg) for all exposures (48 h, 72 h, 96 h, and 120 h).

No significant statistical differences were observed in substrate consumption across the five groups of carbon sources (carbohydrates, carboxylic acids, polymers, amino acids, and amines/amides) in treatments with varying concentrations of ceftriaxone nor according to the microbiota source (soil with or without earthworms and earthworm gut microbiota) after 120 h (Figure 6). However, Figure 7 depicts a heat map displaying individual data for each of the 31 carbon sources, revealing variations in their consumption among treatments at different ceftriaxone concentrations, particularly when comparing the microbiota source (soil with and without earthworms and earthworm gut microbiota). The results of the PERMANOVA analysis confirm that there were no significant differences between treatments with and without earthworms. However, the analysis did reveal that the soil microbiota, regardless of the presence of earthworms, differed from the intestinal microbiota of earthworms (Tables 3 and 4).

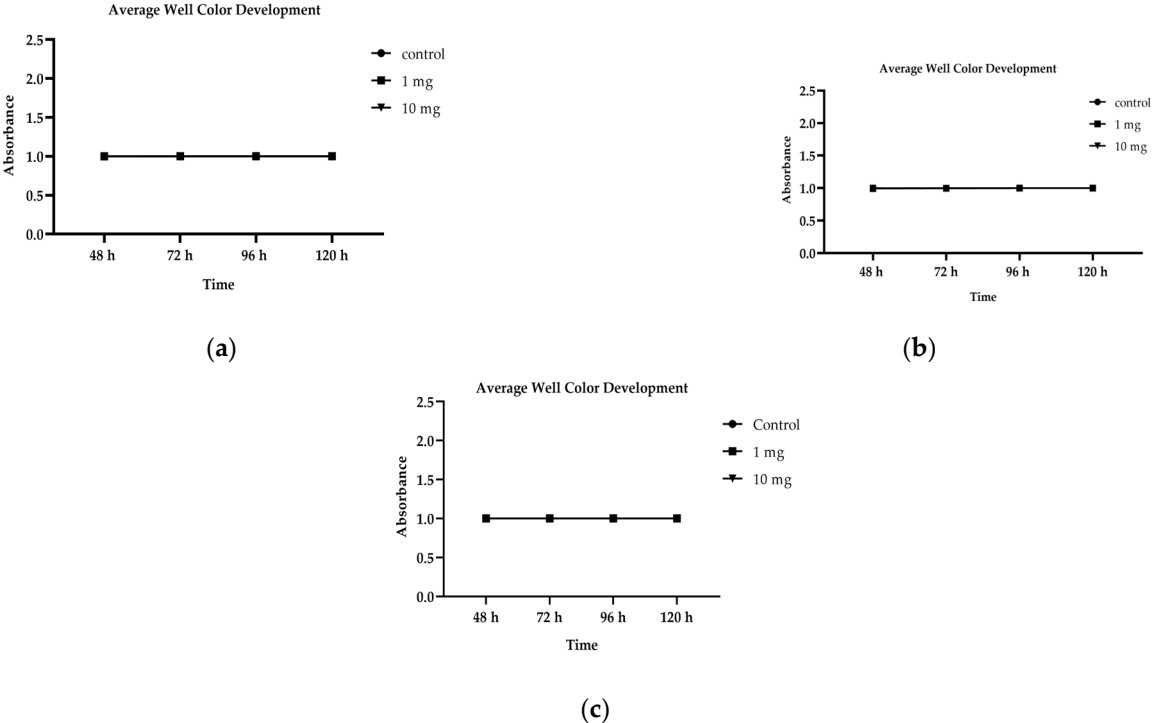

**Figure 2.** Average well color development of (**a**) soil, (**b**) soil with earthworms, and (**c**) earthworm gut under the three treatments (control, 1 mg, and 10 mg) for all exposures (48 h, 72 h, 96 h, and 120 h).

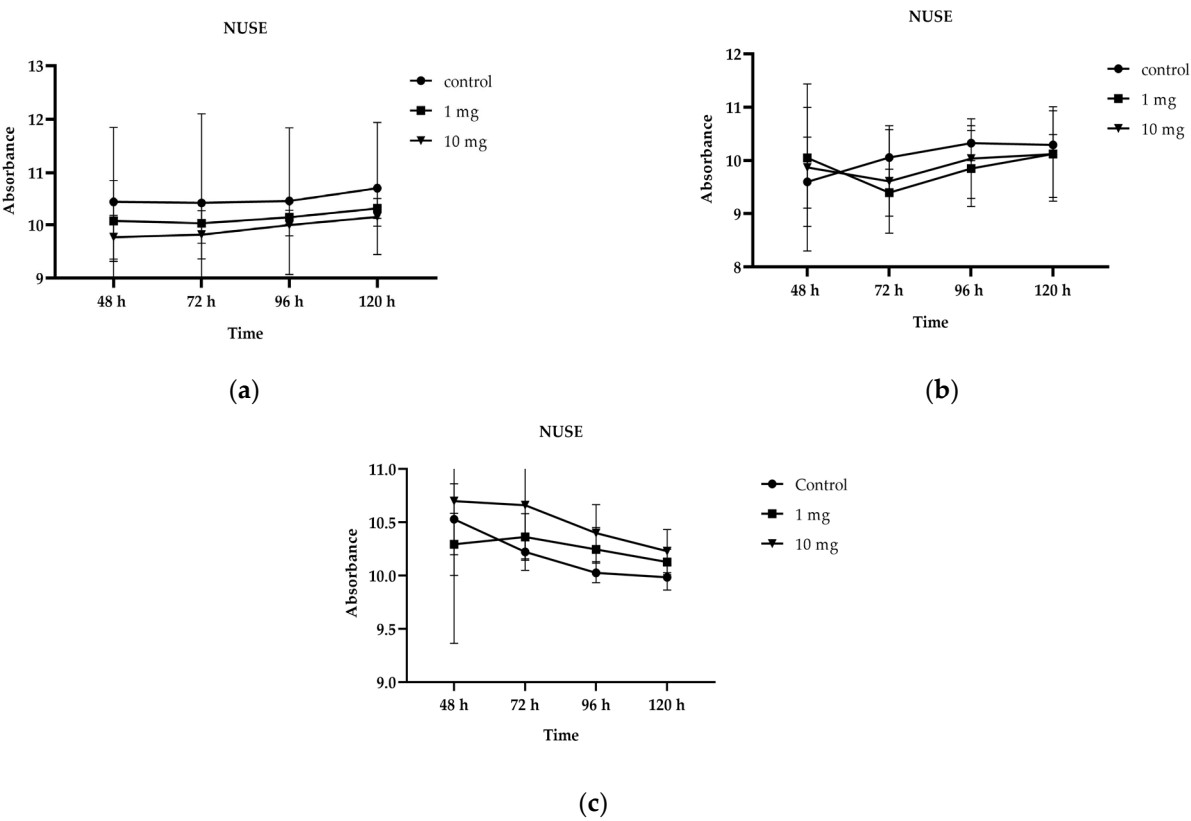

**Figure 3.** NUSE of (**a**) soil, (**b**) soil with earthworms, and (**c**) earthworm gut under the three treatments (control, 1 mg, and 10 mg) for all exposures (48 h, 72 h, 96 h, and 120 h).

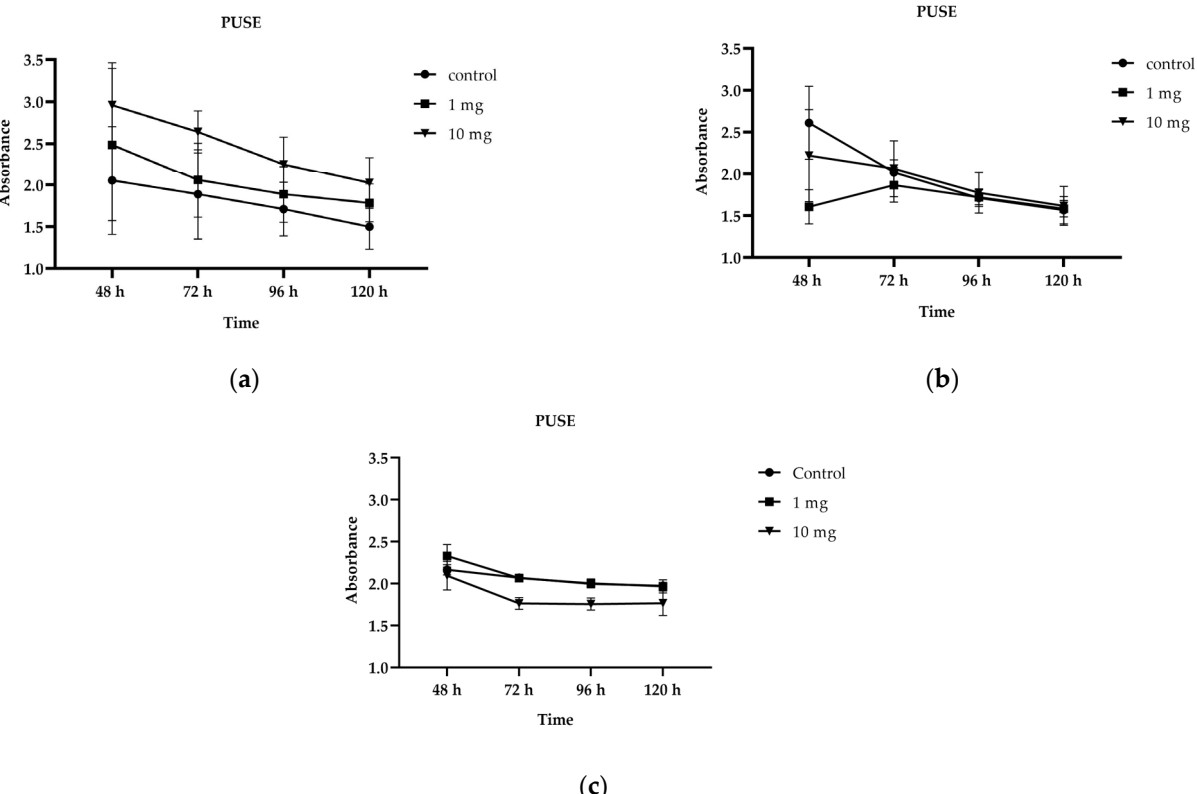

**Figure 4.** PUSE of (**a**) soil, (**b**) soil with earthworms, and (**c**) earthworm gut under the three treatments (control, 1 mg, and 10 mg) for all exposures (48 h, 72 h, 96 h, and 120 h).

**Table 2.** Results of the 2-way ANOVA test of the groups and treatments for the different metabolic profile parameters evaluated using Biolog EcoPlate™.

| Source of Variation | | df | Mean Square | F | p-Value |
|---|---|---|---|---|---|
| AWCD | Interaction | 4 | 0.000006333 | 0.57 | 0.6878 |
| | Groups | 2 | 0.000006333 | 0.57 | 0.5754 |
| | Treatment | 2 | 0.00001633 | 1.47 | 0.2562 |
| | Residual | 18 | 0.00001111 | | |
| Shannon index (H) | Interaction | 4 | 0.001832 | 0.2867 | 0.8827 |
| | Groups | 2 | 0.4749 | 74.32 | 0.0001 |
| | Treatment | 2 | 0.00171 | 0.2677 | 0.7681 |
| | Residual | 18 | 0.006389 | | |
| NUSE | Interaction | 4 | 1.431 | 0.3705 | 0.8265 |
| | Groups | 2 | 3.256 | 0.8428 | 0.4468 |
| | Treatment | 2 | 0.8235 | 0.2132 | 0.8100 |
| | Residual | 18 | 3.863 | | |
| PUSE | Interaction | 4 | 1.774 | 2.906 | 0.0511 |
| | Groups | 2 | 1.093 | 1.79 | 0.1954 |
| | Treatment | 2 | 0.469 | 0.7681 | 0.4785 |
| | Residual | 18 | 0.6106 | | |

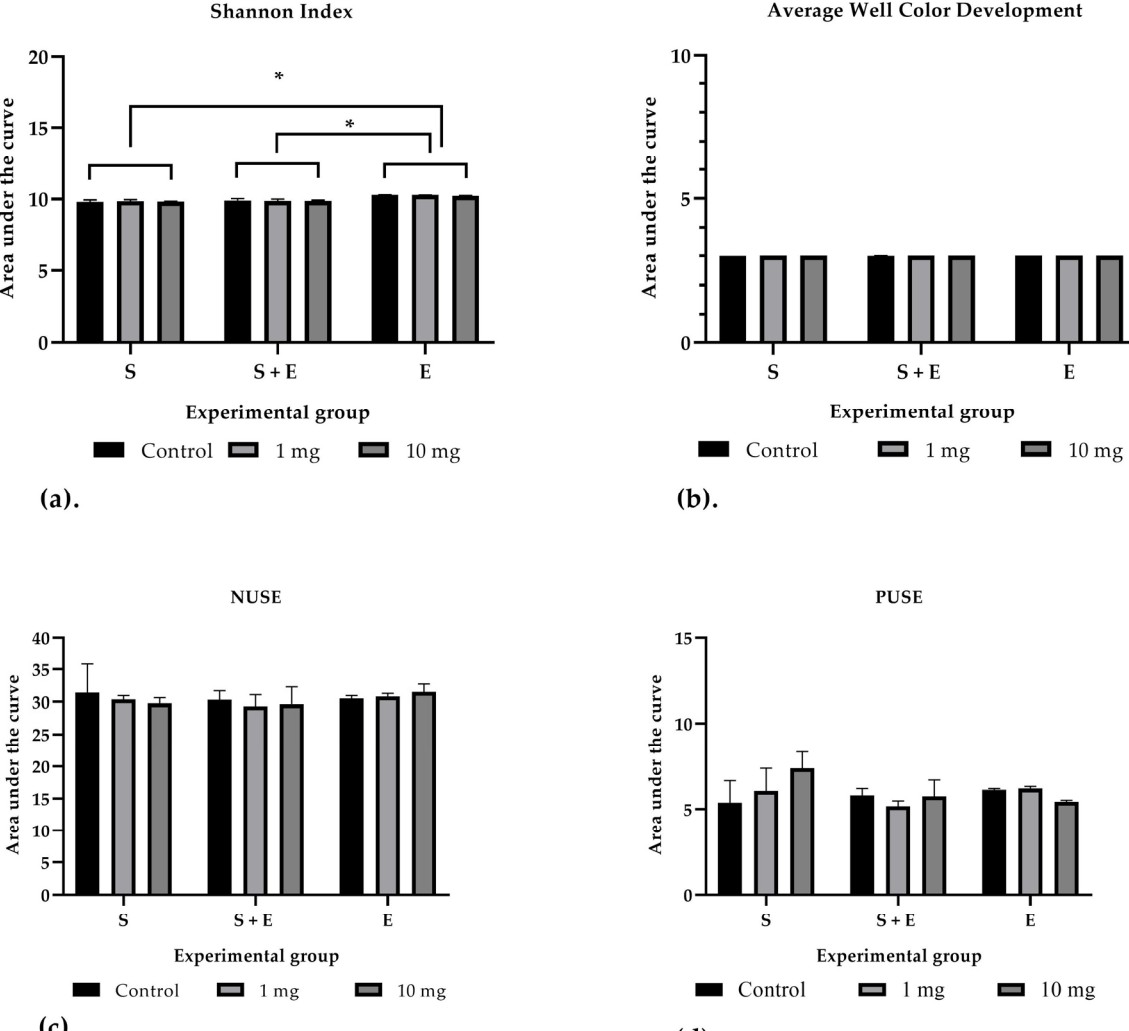

**Figure 5.** Results of the two-way ANOVA test of the area under the curve of the parameters (**a**) Shannon index, (**b**) average well color development, (**c**) NUSE, and (**d**) PUSE in the different experimental groups (soil (S), soil with earthworms (S + E), and earthworm intestine (E)). * $p < 0.05$.

**Table 3.** Results of the 2-way PERMANOVA test on the differences in carbon source consumption among treatments and groups.

| Source of Variation | | *df* | Mean Square | *F* | *p*-Value |
|---|---|---|---|---|---|
| Treatment | 0.009718 | 2 | 0.0048589 | 1.0943 | 0.3353 |
| Group | 0.095721 | 2 | 0.047861 | 10.779 | 0.0001 |
| Interaction | 0.021498 | 4 | 0.0053746 | 1.2104 | 0.2664 |
| Residual | 0.079926 | 18 | 0.0044404 | | |
| Total | 0.20686 | 26 | | | |

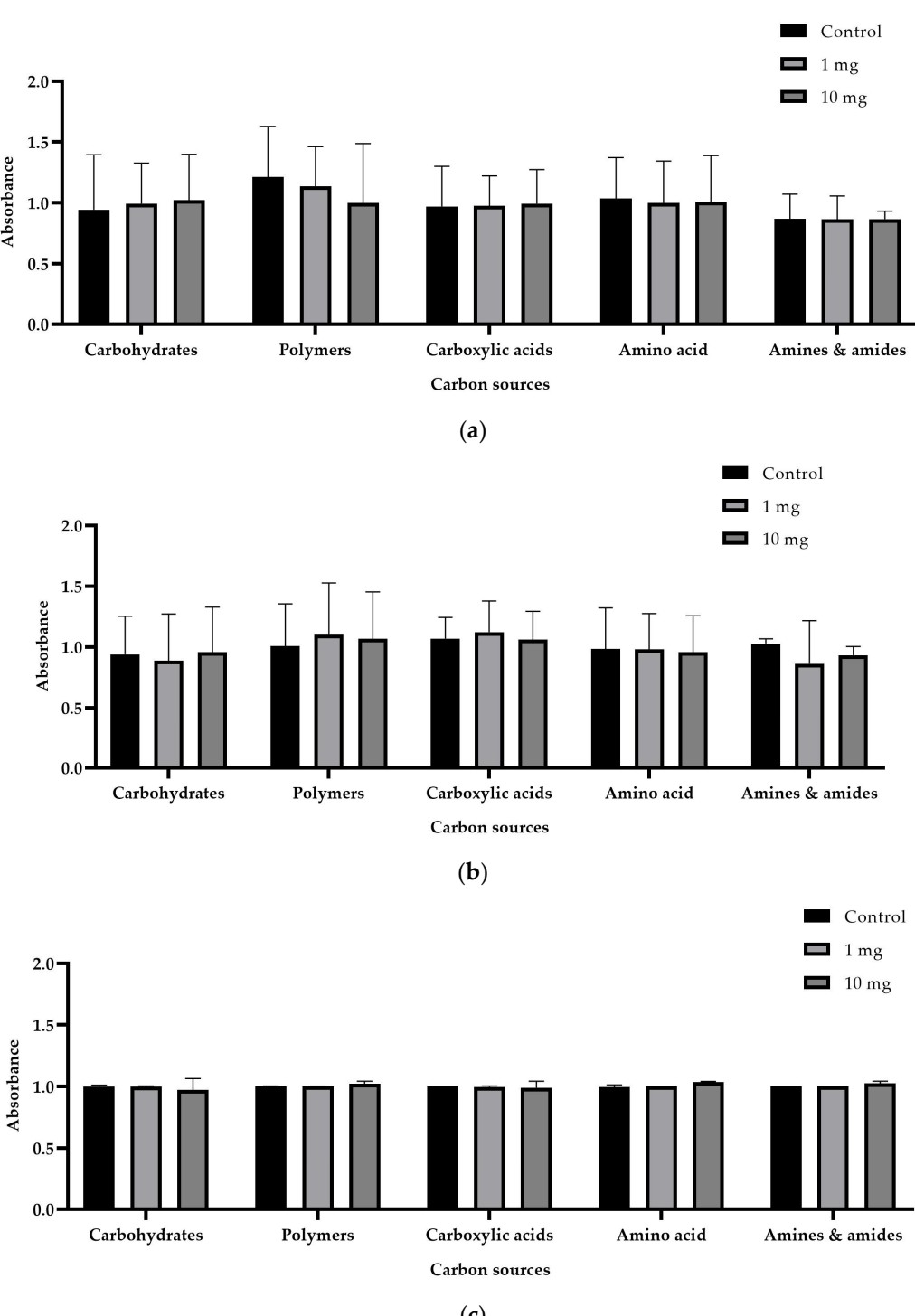

**Figure 6.** This figure illustrates the utilization of different substrate types by microbial communities in (**a**) soil, (**b**) soil with earthworms, and (**c**) earthworm guts across three different treatments (control, 1 mg/kg, and 10 mg/kg) at the 120 h time point.

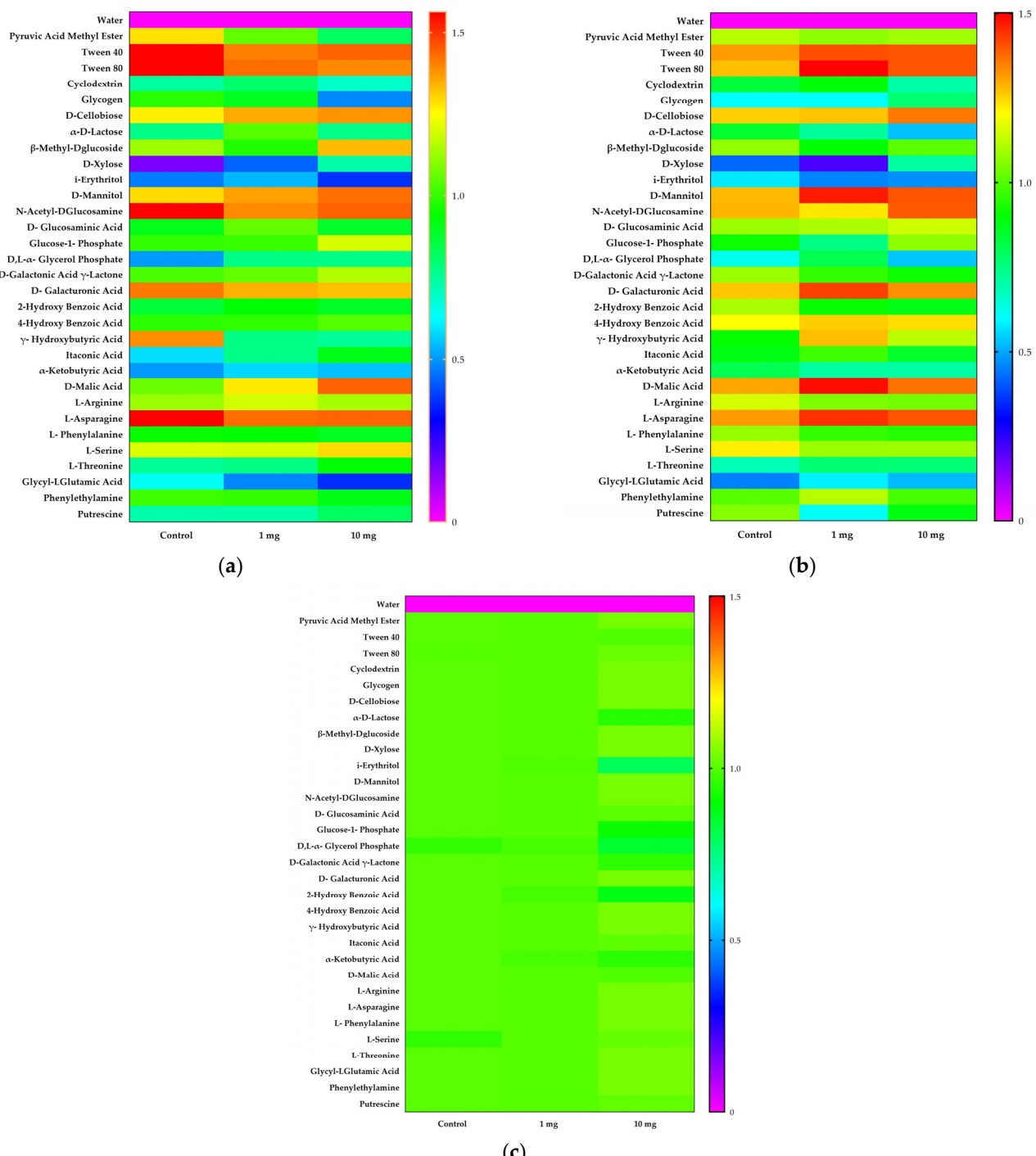

**Figure 7.** Heatmap generated in the Graph Pad Prism program showing the use of the 31 carbon sources in the (**a**) soil, (**b**) soil with earthworms, and (**c**) earthworm gut groups under the three treatments (control, 1 mg/kg, and 10 mg/kg) at the 120 h reading.

**Table 4.** Results of the one-way PERMANOVA test, Bonferroni-corrected *p*-value demonstrating the differences between the groups soil x earthworm gut and soil with earthworms x earthworm gut.

| | Soil | Soil with Earthworms | Earthworms Gut |
|---|---|---|---|
| Soil | | 0.33 | 0.0003 |
| Soil with earthworms | 0.33 | | 0.0003 |
| Earthworms gut | 0.0003 | 0.0003 | |

## 4. Discussion

There is global concern regarding antibiotics, including their excessive and inappropriate use and disposal. These issues have impacts on various organisms, plants, and soil microbiota, and contribute to antibiotic resistance [4]. However, many countries lack specific laws regulating antibiotic levels in the environment [45]. The data from this study suggest that ceftriaxone concentrations in the soil have a minimal influence on the functional profile of soil microbiota and the intestinal microbiota of worms exposed to contaminated soil. Nevertheless, there is a difference in functional diversity between the intestinal microbiota of worms and soil microbiota (with or without worms).

Several hypotheses can be proposed to explain the absence of changes in the microbial functional profile. Firstly, the limited influence of ceftriaxone on the soil bacterial community could be attributed to its low availability in the complex soil compartment, causing it not to exert its antimicrobial activity under the tested conditions. Studies have indicated that ceftriaxone exhibits a strong affinity for minerals, potentially reducing its bioavailability [46]. When it comes to antibiotics in soil, sorption processes can significantly affect the bioavailability of these compounds, leading to a reduction in their antimicrobial activity and limiting their environmental impact. For example, ceftriaxone, as a broad-spectrum antibiotic, exhibits a high affinity for soil minerals, potentially diminishing its availability to bacteria in the environment and consequently reducing its impact on soil microbiota [46].

In addition to sorption mechanisms, a recent study has elucidated the impact of different antibiotic classes on soil microbiota. Ceftriaxone exhibited rapid degradation in soil, completely losing its antimicrobial activity within 19 h. Moreover, the study suggests that unstable antibiotics (such as ceftriaxone) and those with high adsorption rates (such as kanamycin) exert smaller or insignificant influences on the soil bacterial community compared with other antibiotic classes [47]. Furthermore, in terms of antibiotic degradability, it is plausible that ceftriaxone was degraded by bacteria present in the soil. Mardani et al. [48] demonstrated that genetically modified *Pseudomonas putida* bacteria were capable of degrading 69.53% of ceftriaxone in soil. The study highlights the potential efficacy of catechol 2,3-dioxygenase-producing microorganisms in effectively degrading ceftriaxone within a complex matrix such as soil.

One point worth mentioning is the absence of differences in the functional profile of the soil microbiota in the presence or absence of earthworms. A recent review highlighted the variability in microbial abundance in soils in the presence of epigeic earthworms *Eisenia andrei*. Most of the literature indicates that earthworms are capable of either increasing or decreasing microbial biomass, while less frequent studies describe no significant change in microorganism numbers [49]. The literature indicates that soil microbiota can be influenced by the presence of earthworms and that they can contribute to the decontamination and bioremediation of chemically polluted soils, including those contaminated with antibiotics [29]. A study by Pu et al. [28] demonstrated that earthworm activity altered the composition of the microbial community in ciprofloxacin-contaminated soil. The authors also concluded that the gut of earthworms plays a crucial role in ciprofloxacin removal and could be a potential strategy for mitigating antibiotic pollution in soil. However, it is important to consider the concentration of the antibiotic in soil, as pronounced effects of ceftriaxone might be observed at higher concentrations.

The study conducted by Xia et al. [50] suggests that low concentrations of antibiotics in the soil may not have a detrimental impact on the soil microbial community, and earthworms may even derive benefits from the presence of these compounds. Another study focusing on soils contaminated with polycyclic aromatic hydrocarbons highlighted that earthworm activity altered the composition of the soil microbiota but did not affect the consumption of carbon substrates [51]. Thus, it is possible that structural community changes mediated by chemicals present in the soil may not affect the functional profile of this microbiota due to resilience or functional redundancy. The concentration of ceftriaxone in the soil appears to have been unable to affect the pattern of substrate consumption by the earthworm gut microbiota. There was little similarity in the functional profile of the soil

microbiota in the presence of earthworms. However, in the gut contents of these organisms, that was a bigger consumption of the different substrates analyzed (Figure 7c). Therefore, studies such as Wang et al.'s [23] indicate that the functional diversity of earthworm gut microbiota has a direct relationship with soil microbiota. Another study conducted with a different soil organism also demonstrated a direct correlation between soil microbiota and the gut microbiota of terrestrial isopods. Volcão et al. [37] revealed that the gut microbiota of the terrestrial isopod *Baloniscus selowii*, exposed to soil contaminated with antimicrobial agents (chlorhexidine and triclosan), exhibited functional diversity remarkably similar to that of soil microbiota under the same treatments. The authors attributed these findings to the contaminated soil microbiota being the primary source of microorganisms for the digestive tract of the terrestrial isopod.

Some researchers have pointed out limitations to the use of Biolog EcoPlate carbon, such as the bias of tetrazolium dye, as not all bacteria are capable of reducing it. Thus, the plates might not provide a comprehensive view of the microbial community [52]. However, Biolog EcoPlate is a valuable tool for assessing the functional profile of soil microbiota and the gut microbiota of earthworms in antibiotic-contaminated soil. The findings of our study suggest that even at concentrations as high as 1 mg/kg and 10 mg/kg of ceftriaxone, the physiological profile of the soil microbiota remained largely unchanged. Despite the recognized importance of earthworms in maintaining soil quality, our results did not demonstrate a significant correlation between alterations in the microbial community and the presence of earthworms. These observations underscore the intricate nature of soil microbiota and emphasize the need for further research to fully comprehend the underlying mechanisms of these interactions. Future investigations should analyze the short-term, medium-term, and long-term impacts of ceftriaxone and other cephalosporins on soil microbiota, as these studies are crucial to the development of effective management strategies aimed at promoting biodiversity and safeguarding soil health and its edaphic components.

## 5. Conclusions

The findings of our study indicate that even when using high concentrations of ceftriaxone (1 mg/kg and 10 mg/kg), the soil microbiota remained mostly unaffected. Although earthworms are known to play a vital role in maintaining soil quality, our results did not show a significant relationship between changes in the microbial community and the presence of earthworms. These observations highlight the intricate nature of soil microbiota and emphasize the need for further research to fully understand the underlying mechanisms involved in these interactions.

**Author Contributions:** Conceptualization, L.d.S.F., F.M.R.d.S.J. and D.F.R.; methodology, L.d.S.F., R.d.L.B., A.d.S.B. and L.M.V.; software, L.d.S.F. and R.d.L.B.; formal analysis, L.d.S.F., R.d.L.B. and L.M.V.; investigation, L.d.S.F., F.M.R.d.S.J. and D.F.R.; resources, L.d.S.F., F.M.R.d.S.J. and D.F.R.; data curation, L.d.S.F., F.M.R.d.S.J. and D.F.R.; writing—original draft preparation, L.d.S.F., F.M.R.d.S.J. and D.F.R.; writing—review and editing, L.d.S.F., R.d.L.B., A.d.S.B., L.M.V., F.M.R.d.S.J. and D.F.R.; visualization, L.d.S.F., F.M.R.d.S.J. and D.F.R.; supervision, F.M.R.d.S.J. and D.F.R.; project administration, F.M.R.d.S.J. and D.F.R.; funding acquisition, F.M.R.d.S.J. All authors have read and agreed to the published version of the manuscript.

**Funding:** This research was funded by Coordenação de Aperfeiçoamento de Pessoal de Nível Superior (CAPES), Brasil, Finance Code 001, and by Conselho Nacional de Desenvolvimento Científico e Tecnológico (CNPq) (Research Productivity Grant 310856/2020-5 to Flavio Manoel Rodrigues da Silva Júnior and Research Productivity Grant 306806/2022-3 to Daniela Fernandes Ramos).

**Institutional Review Board Statement:** Not applicable.

**Informed Consent Statement:** Not applicable.

**Data Availability Statement:** Not applicable.

**Acknowledgments:** The authors thank Coordenação de Aperfeiçoamento de Pessoal Superior (CAPES) and Conselho Nacional de Desenvolvimento Científico e Tecnológico (CNPq) for granting the scholarship.

**Conflicts of Interest:** The authors declare no conflict of interest.

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
