# Peer review of "Assessment of the Impact of Ceftriaxone on the Functional Profile of Soil Microbiota Using Biolog EcoPlateTM"

_soilsystems, doi:10.3390/soilsystems7020055_

Round 1

Reviewer 1 Report

The manuscript has tried to interpret the role of modern innovation viz Biolog Ecoplate in screening of microbial activity but it suffers from some major lacunas:

1. Author should clearly state the background of research in a very lucid english. The manuscript should include the significance of the research.

2. In the abstract section, the following statement is  

"The functional profile of the 28 microbiota of the soils with and without earthworms was similar, but differed from the profile of 29 the intestinal microbiota of the earthworms." 

I think this research plan is insignificant. The author can think some other aspects in field of microbial ecology with wide range of broad spectrum antibiotics which will reveal some new ideas in vermitechnology.

3. Manuscript should be written in lucid english.

English should be changed through out the length and breadth of chapter.

Author Response

Review #1:

The manuscript has tried to interpret the role of modern innovation viz Biolog Ecoplate in screening of microbial activity but it suffers from some major lacunas:

  1. Author should clearly state the background of research in a very lucid english. The manuscript should include the significance of the research.

A: Thank you very much. We have changed the last paragraph of the introduction to reflect the importance of the study and the background.

  1. In the abstract section, the following statement is  

"The functional profile of the 28 microbiota of the soils with and without earthworms was similar, but differed from the profile of 29 the intestinal microbiota of the earthworms." 

I think this research plan is insignificant. The author can think some other aspects in field of microbial ecology with wide range of broad spectrum antibiotics which will reveal some new ideas in vermitechnology.

A: This information was included in the abstract because previous studies report a similarity between the intestinal microbiota of terrestrial invertebrates and the soil microbiota. We included a PERMANOVA analysis (at the suggestion of reviewer #2) to estimate the similarity of treatments.

  1. Manuscript should be written in lucid english.

A: Thank you very much. English has been precisely corrected.

Reviewer 2 Report

Introduction:

 Excessive use of short paragraphs; please consider joining some of them

 Page 3, Line 89: dehydrogenase, phosphatase and urease are enzymes and not techniques

Materials and Methods:

Page 4, Lines 155-156: the categories of the substrates are different from the categories presented in the table 1. They also differ from the table legend

Results:

Page 7, Lines 222-228: it is noteworthy that the antibiotic treatments displayed a slightly higher diversity of the substrates in the soil without the earthworms. Was this difference significant? Also, what are the two other parameters not shown? at least they should be mentioned

 Page 9, 260-261: authors just refer to heat maps qualitatively. They should have performed some kind of multivariate analysis to test differences between the treatments. I highly encourage the use of PERMANOVA, PERMDISP and SIMPER analyses, as they could help more suitable sustain the conclusions.

 Results of ANOVA are not properly displayed. Please provide the tables

Discussion:

 Page 11, Line 293: “functional diversity” is a concept that has not been properly introduced before

Page 12, Lines 312-315: this sentence is repeated

Pages 11-12, Lines 296-314: the statements presented are in disagreement with the very reasonings to conduct the study

Page 12, Line 325: please pay attention to the highlighting of species names (here and other parts of the manuscript)

 Limitations of the ecoplate method shoud be pointed, as they could have impacted the conclusions. For instance, this method only captures part of the culturable microbial community and can only be assumed to be a function of the total community physiological capabilities. Also, working with soil samples and ecoplates is tricky because of the typical high amounts of organic matter of this matrix. This can mask the results, because microbes can grow on these substrates instead of the commercial substrates (high false positive rates). Moreover, other organisms such as archaea and fungi could have been competitively benefited from the antibiotic treatments, and they are known to be able to grow in ecoplates as well.

Tables and Figures:

Figure 1: please provide error bars (preferentially standard deviation)

Figure 2: rescale 2b and 2d. The Y axis is too large for the bars

Author Response

Review #2

Introduction:

 Excessive use of short paragraphs; please consider joining some of them

A: Many thanks for all the suggestions. The improvement of English was able to solve this problem.

 Page 3, Line 89: dehydrogenase, phosphatase and urease are enzymes and not techniques

 A: The error has been fixed

Materials and Methods:

Page 4, Lines 155-156: the categories of the substrates are different from the categories presented in the table 1. They also differ from the table legend

 A: The error has been fixed

Results:

Page 7, Lines 222-228: it is noteworthy that the antibiotic treatments displayed a slightly higher diversity of the substrates in the soil without the earthworms. Was this difference significant? Also, what are the two other parameters not shown? at least they should be mentioned

A: The suggested data were presented in figures 2, 3 and 4

 Page 9, 260-261: authors just refer to heat maps qualitatively. They should have performed some kind of multivariate analysis to test differences between the treatments. I highly encourage the use of PERMANOVA, PERMDISP and SIMPER analyses, as they could help more suitable sustain the conclusions.

A: Thanks for the sugestion. We included the suggested analysis (Tables 3 and 4)

 Results of ANOVA are not properly displayed. Please provide the tables

A: Thanks for the sugestion. We include the data in Table 2.

Discussion:

 Page 11, Line 293: “functional diversity” is a concept that has not been properly introduced before

A: We use functional diversity as a synonym for physiological diversity. We include this in the material and methods section (line 203)

Page 12, Lines 312-315: this sentence is repeated

 A: The error has been fixed

Pages 11-12, Lines 296-314: the statements presented are in disagreement with the very reasonings to conduct the study

A: We agree with the reviewer's comment. Based on our initial hypothesis, an impact of the antibiotic on the soil microbiota was expected. At reviewer #1's suggestion, we include this background in the introduction. However, our findings showed no impact on the tested conditions. These two paragraphs in the discussion list arguments about the absence of differences.

Page 12, Line 325: please pay attention to the highlighting of species names (here and other parts of the manuscript)

 A: The error has been fixed

 Limitations of the ecoplate method shoud be pointed, as they could have impacted the conclusions. For instance, this method only captures part of the culturable microbial community and can only be assumed to be a function of the total community physiological capabilities. Also, working with soil samples and ecoplates is tricky because of the typical high amounts of organic matter of this matrix. This can mask the results, because microbes can grow on these substrates instead of the commercial substrates (high false positive rates). Moreover, other organisms such as archaea and fungi could have been competitively benefited from the antibiotic treatments, and they are known to be able to grow in ecoplates as well.

Tables and Figures:

Figure 1: please provide error bars (preferentially standard deviation)

 A: The error has been fixed

Figure 2: rescale 2b and 2d. The Y axis is too large for the bars

 A: The error has been fixed

Reviewer 3 Report

Soilsystems-2346067 provide a help for understanding the the impact of ceftriaxone on the functional profile of the soil microbiota using the Biolog Ecoplate. The content of the work is meaningful. However, the manuscript has great problems in experiment design, writing style and result presentation. Thus, I suggested that the manuscript need to be major revised by authors.

1. It is suggested that the manuscript needs language editing. The research described in this manuscript is made more difficult to understand technically by the difficult-to-understand English language usage. It is noted that your manuscript needs careful editing by someone with expertise in technical English editing paying particular attention to English grammar, spelling, and sentence structure so that the goals and results of the study are clear to the reader.

2. Line 16: Background should be bolded.

3. Abstract section: The conclusion should be added.

4. Keywords should be reduced to five.

5. The logic of introduction needs to be further improved. Some of the paragraphs are long and short, which is very messy.

6. Materials and Methods: Secondary headings should be added.

1. Experimental design: Experimental design was not well constructed. Why chose the two different Ceftriaxone concentrations, what is the basis? This is a big problem?

7. Results: Secondary headings should be added.

8. Conclusions should be further streamlined and improved.

Soilsystems-2346067 provide a help for understanding the the impact of ceftriaxone on the functional profile of the soil microbiota using the Biolog Ecoplate. The content of the work is meaningful. However, the manuscript has great problems in experiment design, writing style and result presentation. Thus, I suggested that the manuscript need to be major revised by authors.

1. It is suggested that the manuscript needs language editing. The research described in this manuscript is made more difficult to understand technically by the difficult-to-understand English language usage. It is noted that your manuscript needs careful editing by someone with expertise in technical English editing paying particular attention to English grammar, spelling, and sentence structure so that the goals and results of the study are clear to the reader.

2. Line 16: Background should be bolded.

3. Abstract section: The conclusion should be added.

4. Keywords should be reduced to five.

5. The logic of introduction needs to be further improved. Some of the paragraphs are long and short, which is very messy.

6. Materials and Methods: Secondary headings should be added.

1. Experimental design: Experimental design was not well constructed. Why chose the two different Ceftriaxone concentrations, what is the basis? This is a big problem?

7. Results: Secondary headings should be added.

8. Conclusions should be further streamlined and improved.

Author Response

Review #3

Soilsystems-2346067 provide a help for understanding the the impact of ceftriaxone on the functional profile of the soil microbiota using the Biolog Ecoplate. The content of the work is meaningful. However, the manuscript has great problems in experiment design, writing style and result presentation. Thus, I suggested that the manuscript need to be major revised by authors.

  1. It is suggested that the manuscript needs language editing. The research described in this manuscript is made more difficult to understand technically by the difficult-to-understand English language usage. It is noted that your manuscript needs careful editing by someone with expertise in technical English editing paying particular attention to English grammar, spelling, and sentence structure so that the goals and results of the study are clear to the reader.

A: Thanks for the suggestion. The request has been granted.

  1. Line 16: ‘Background’ should be bolded.

A: Thanks for the suggestion. The request has been granted.

  1. Abstract section: The conclusion should be added.

A: Thanks for the suggestion. The request has been granted.

  1. Keywords should be reduced to five.

A: Thanks for the suggestion. The request has been granted.

  1. The logic of introduction needs to be further improved. Some of the paragraphs are long and short, which is very messy.

A: This suggestion was also made by reviewers #1 and #2. English improvement has fixed this problem.

  1. Materials and Methods: Secondary headings should be added.

A: Thanks for the suggestion. The request has been granted.

Experimental design: Experimental design was not well constructed. Why chose the two different Ceftriaxone concentrations, what is the basis? This is a big problem?

A: We chose to use the same concentrations used in the study by Orlewska and colleagues [38] that had already demonstrated an impact on the soil microbiota and that are compatible with environmental concentrations of antibiotics. This is described in section 2.2

  1. Results: Secondary headings should be added.

A: Thanks for the suggestion. The request has been granted.

  1. Conclusions should be further streamlined and improved.

A: Thanks for the suggestion. The request has been granted.

Round 2

Reviewer 3 Report

I have no comment.

I have no comment.